# Room temperature valley polarization via spin selective charge transfer

Shreetu Shrestha [1], Mingxing Li[1], Suji Park[1], Xiao Tong[1], Donald DiMarzio [2] & Mircea Cotlet [1]

The two degenerate valleys in transition metal dichalcogenides can be used to store and process information for quantum information science and technology. A major challenge is maintaining valley polarization at room temperature where phonon-induced intervalley scattering is prominent. Here we demonstrate room temperature valley polarization in heterostructures of monolayer $MoS_2$ and naphthylethylammine based one-dimensional chiral lead halide perovskite. By optically exciting the heterostructures with linearly polarized light close to resonance and measuring the helicity resolved photoluminescence, we obtain a degree of polarization of up to −7% and 8% in $MoS_2$/right-handed (R-(+)-) and left-handed (S-(-)-) 1-(1-naphthyl)ethylammonium lead iodide perovskite, respectively. We attribute this to spin selective charge transfer from $MoS_2$ to the chiral perovskites, where the perovskites act as a spin filter due to their chiral nature. Our study provides a simple, yet robust route to obtain room temperature valley polarization, paving the way for practical valleytronics devices.

Besides an electron's charge and spin, the valley it occupies in momentum space can also be used to process and store information for classical or quantum computing. As the valleys are often well separated in momentum space, the valley index is robust against scattering[1] and offers a reliable new degree of freedom. Two-dimensional (2D) materials with a honeycomb lattice such as transition metal dichalcogenides (TMDs) have two degenerate valleys, K and K', at the corners of their hexagonal Brillion zone (Fig. 1a) and are great candidates for valleytronics. Monolayers of group VI of TMDs in 2H phase have a direct bandgap at the K point and a broken crystal inversion symmetry ($D^1_{3h}$) which leads to valley-dependent optical selection rules[2]. Moreover, strong spin–orbit coupling due to the heavy metal atoms splits the electronic band by hundreds of meV[3]. Since time reversal symmetry requires that the spin splitting at the two valleys is opposite, spin and valley are coupled[1] or locked. Therefore, to change an electron's valley, a simultaneous flip in its spin is required which makes valley index in monolayer TMDs robust.

Selectively populating a specific valley, or valley polarization which can be considered as writing or encoding data, is the first step towards valleytronics applications. The most common route used to obtain valley polarization in TMDs is via optical pumping with a circularly polarized light. Due to the valley-dependent optical selection rules, K and K' valleys can be selectively populated with left (σ+) and right (σ−) circularly polarized light, respectively. Similarly, valley polarization can be read out by measuring the σ+ and σ− components of the emitted photoluminescence (PL)[4–8]. The valley population in TMDs can be further manipulated using strain[9], magnetic field[10,11], or by adding plasmonic metasurfaces[12]. Moreover, taking advantage of the layered nature of TMDs, enhanced valley polarization due to charge transfer and other proximity effects in heterostructures consisting of TMDs[13,14], other 2D materials[15] and magnetic materials[16] has also been demonstrated.

Valley polarization was recently demonstrated in heterostructures combining TMDs and 2D chiral lead halide perovskites[17]. In these heterostructures, a chirality induced spin selectivity effect[18,19] results in chiral perovskites acting as spin filters, preferentially injecting charges with one spin, depending on the handedness of the perovskite material. Since spin and valley in monolayer TMDs are coupled,

[1]Center for Functional Nanomaterials, Brookhaven National Laboratory, Upton, NY 11973, USA. [2]Northrop Grumman Corporation, One Space Park, Redondo Beach, CA 90278, USA. ✉e-mail: sshrestha@bnl.gov; cotlet@bnl.gov

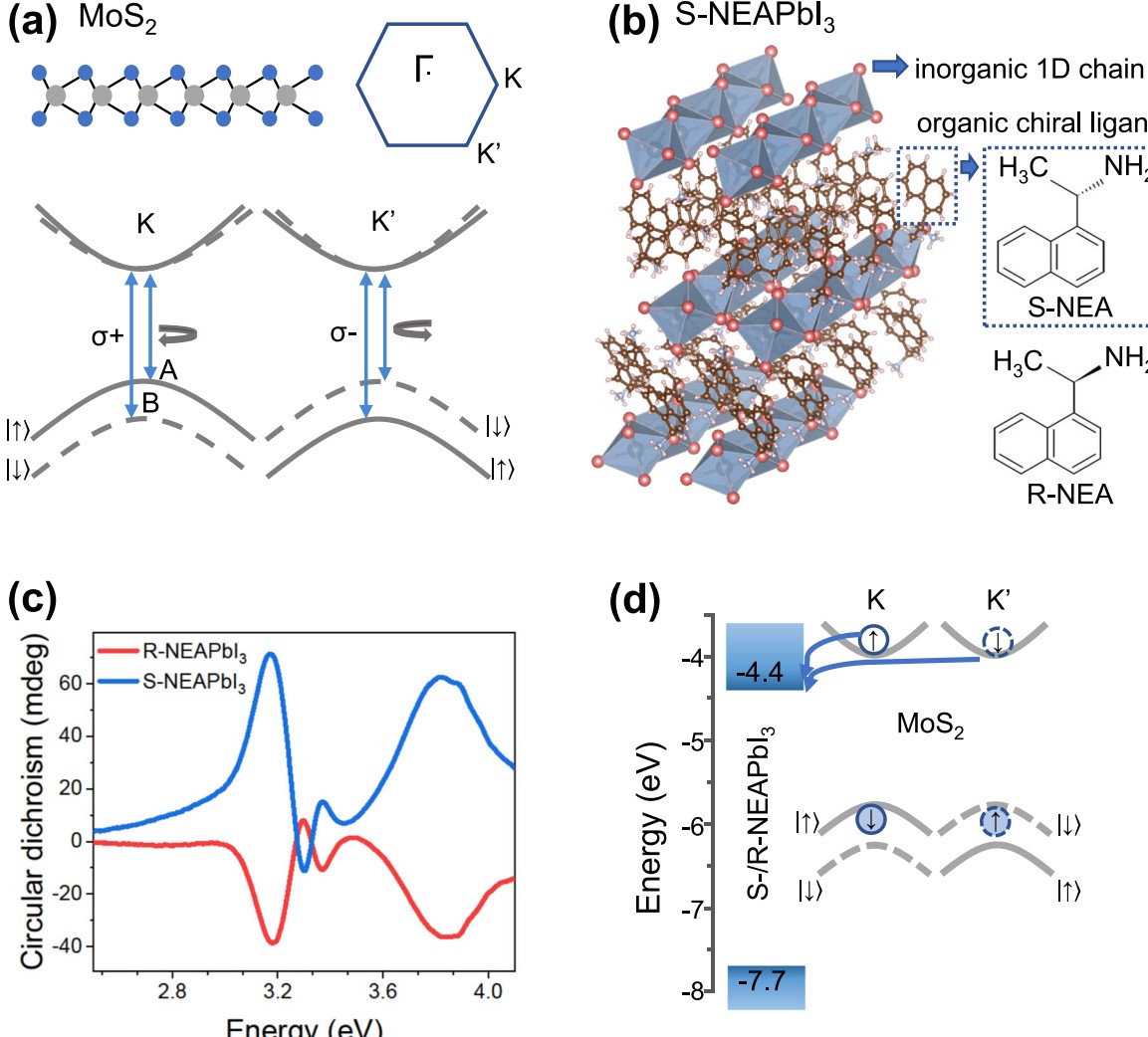

**Fig. 1 | Heterostructure of monolayer MoS₂ and 1D chiral perovskite. a** side view of the crystal structure of a monolayer MoS₂ showing the broken crystal inversion symmetry (Mo and S atoms shown in gray and blue, respectively). The hexagonal Brillion zone has K and K' points at the corners. The electronic band structure at these points (bottom) has a direct bandgap with spin split valence band. A and B are exciton transitions from the conduction band to the upper and lower split valence bands, respectively. Due to optical selection rules, only K (K') valley emits or absorbs σ+ (σ−). **b** crystal structure of chiral perovskite (S-(-)-) 1-(1-naphthyl)ethylammonium lead iodide (S-NEAPbI₃)[24]. The inorganic 1D lead halide chain (Pb and I atoms shown in gray and red) is surrounded by chiral organic ligands. The dotted blue box shows the molecular structure of the chiral organic ligand S-1-(1-naphthyl) ethylamine (S-NEA). The molecular structure of the enantiomer R-1-(1-naphthyl) ethylamine (R-NEA) is also shown. **c** Circular dichroism spectra of thin films of R-NEAPbI₃ (red) and S-NEAPbI₃ (blue). **d** Energy band alignment of the hetero-structure components. The electronic bands for S-NEAPbI₃ are derived from ultraviolet photoelectron spectroscopy and UV-vis measurements. The energy is relative to the vacuum level. The bands for monolayer MoS₂ are taken from literature[32,33] and the valence band splitting is exaggerated for clarity. Electrons and holes are depicted as empty and filled circles, respectively.

the valley population is therefore controlled via selective spin injection. This elegant, yet simple way to control valley polarization does not require circularly polarized light, magnetic field, or complex device structure. However, the phenomenon was observed only at low temperature (78 K) and steadily decreased as the temperature increased, leaving no polarization at room temperature. In fact, the majority of studies mentioned here so far have been performed only at low temperatures[4–8,10,11,13,14,16], as depolarization due to intervalley scattering becomes prominent at higher temperature[17,20]. Requirement of cryogenic temperatures makes these systems expensive, bulky, and complex. Achieving robust valley polarization in TMDs at room temperature is a challenge which needs to be addressed for practical applications of TMD-based valleytronics devices in quantum information science and technology.

Here we demonstrate room temperature valley polarization in heterostructures of monolayer MoS₂ and one-dimensional (1D) chiral

lead halide perovskite R-(+)- or S-(-)-1-(1-naphthyl)ethylammonium lead iodide (R- or S-NEAPbI₃). We obtain a degree of polarization of up to 8% by exciting MoS₂ with a linearly polarized laser with energy close to A-exciton resonance, at ambient conditions. We attribute the ability to observe room temperature valley polarization to a stronger chirality of the 1D perovskite system versus earlier reported 2D perovskite together with resonant excitation conditions employed here. Our results demonstrate a simple and robust method to control room temperature valley polarization in TMDs, thus paving the way for practical valleytronics devices which could change the way we store and process information using conventional charge and spin-based electronic devices.

## Results
We begin by synthesizing single crystals of R/S-NEAPbI₃ chiral lead halide perovskites using a slow cooling method[21–23]. Powder X-ray

diffraction pattern of the crystals are consistent with literature[22] (Supplementary Fig. 1) and show the absence of impurities. Figure 1b shows the crystal structure of 1D S-NEAPbI$_3$ chiral lead halide perovskite[24]. The hybrid organic-inorganic material consists of inorganic 1D chains of face-sharing lead halide octahedra surrounded by organic chiral ligands S-(-)-1-(1-naphthylamine). The organic chiral ligands form asymmetric hydrogen bonds with the inorganic framework and have been reported to transfer chirality via helical structural distortions[22] and asymmetric electronic interactions[25]. In comparison to 2D chiral perovskites, where chiral ligands are located between 2D lead-halide sheets, the 1D chiral perovskites show a stronger chiroptical activity because the chiral ligands surrounding the lead-halide chains lead to a stronger distortion in the later systems[26,27]. The circular dichroism (CD) spectra of spin cast thin films of R- and S-NEAPbI$_3$ in Fig. 1c show strong CD signals, with peaks appearing at the same energy, but with opposite signs for the two enantiomers. This confirms their preference for absorption of light with opposite helicities. The bisignate signal at about 3.2 eV corresponds to exciton absorption (Supplementary Fig. 2) and is in agreement with literature[26].

Weak van der Waals interactions between the organic ligands allows the chiral perovskite crystals to be mechanically exfoliated into thin flakes with smooth surfaces[28–30] (confirmation by atomic force microscopy (AFM) in Supplementary Fig. 3). To make heterostructures, MoS$_2$ flakes were mechanically exfoliated on Si/SiO$_2$ substrate and monolayers were identified using AFM and Raman microspectroscopy (Supplementary Fig. 4). Next, chiral perovskite flakes roughly 500 nm thick were stacked on top of the monolayer MoS$_2$ using a dry transfer method[31]. Electronic energy bands measured using ultraviolet photoelectron spectroscopy (UPS) show that the chiral perovskites have a valence band maxima at −7.7 eV relative to the vacuum level (Supplementary Fig. 5). Considering a bandgap of 3.3 eV estimated from the UV-Vis absorption data (Supplementary Fig. 2), the conduction band minima is at about −4.4 eV relative to the vacuum level, which provides a type II (staggered) energy band alignment between the perovskite and monolayer MoS$_2$ (Fig. 1d).

## Photoluminescence measurements

We optically excited the heterostructure with a linearly polarized laser of 1.96 eV energy and measured the emitted photoluminescence (PL) in a confocal microscope (see "Methods" for details). As the chiral perovskite has a bandgap of 3.3 eV (Supplementary Fig. 2), we expect negligible absorption of the laser light by this component of the heterostructure. Figure 2a, b shows the reflection and PL maps of a R-NEAPbI$_3$/MoS$_2$ heterostructure, where white and red dash lines outline the monolayer MoS$_2$ and the chiral perovskite flake R-NEAPbI$_3$, respectively. The overlap region of the heterostructure demonstrates a quenched PL signal in comparison to the pristine monolayer MoS$_2$ region. The PL spectra in Fig. 2c shows that both monolayer MoS$_2$ region and the overlap region of the heterostructure have peaks at 1.86 eV, which we relate to the A-exciton optical transition from the conduction band to the upper split valence band in MoS$_2$. However, the PL spectrum from the overlap region is much lower in intensity. We obtained similar results also for the S-NEAPbI$_3$/MoS$_2$ heterostructure which are shown in Fig. 2d–f. Although there is some variation in the intensity of PL emission from point to point on a sample, the PL from the overlap regions is quenched, on average, 50% in comparison to the pristine monolayer region. Looking at the type II energy band alignment of the heterostructure components shown in Fig. 1d, photoinduced electron transfer is expected to occur from the conduction band of the photo-excited monolayer MoS$_2$ to R-NEAPbI$_3$. This photoinduced charge transfer competes with radiative recombination in MoS$_2$, resulting in a quenched PL of MoS$_2$. Emission at low energy from interlayer excitons, which have been reported in TMD/2D perovskite heterostructures[20,34], was absent in the spectral range probed (1.3–2 eV). Therefore, in this work we focus on only MoS$_2$ emission.

The polarization resolved PL signal of monolayer MoS$_2$ (Fig. 2g) shows that the left (σ+) and right (σ−) circularly polarized components of the PL spectrum are identical in peak position, shape, as well as intensity. The R-NEAPbI$_3$/MoS$_2$ heterostructure (Fig. 2h), however, has a higher PL emission peak intensity for σ− (dotted line), while the S-NEAPbI$_3$/MoS$_2$ heterostructure (Fig. 2i) has a higher PL emission peak intensity for σ+ (solid line). Because we optically pump both K and K′ valleys equally with a linearly polarized light, for monolayer MoS$_2$, the PL emitted resulting from radiative recombination from the K and K′ valleys is expected to be similar, as observed here in Fig. 2g. The PL emission differences observed for the heterostructures can be explained taking into account the electronic band structure of MoS$_2$ (Fig. 1a), where spin–orbit coupling splits the valence band maxima into two bands separated by about 150 meV[3]. Excitation with the 1.96 eV laser light promotes electrons only from the upper spin split valence band, associated with the A-exciton, and this populates the conduction band with spin up electrons in the K valley and spin down electrons in the K′ valley. The photo-excited electrons in the conduction band of MoS$_2$ are then driven by the type II (staggered) heterostructure to the conduction band of chiral perovskite (Fig. 1d). However, here the chiral perovskite acts as a spin filter: R-NEAPbI$_3$ preferentially accepts spin up electrons, while S-NEAPbI$_3$ preferentially accepts spin down electrons. This creates a population imbalance in the two valleys, resulting in a higher σ− PL emission intensity in the R-NEAPbI$_3$/MoS$_2$ heterostructures and vice versa in the S-NEAPbI$_3$/MoS$_2$ heterostructures (as shown in insets in Fig. 2h-i). In the overlap region with bilayers MoS$_2$ in Fig. 2e, we observe negligible difference between σ+ and σ− PL emission (Supplementary Fig. 6). This is expected since the inversion symmetry is preserved in bilayers and therefore spin and valley are not coupled[6].

To quantify the chirality of the PL emission, we use the degree of polarization (DOP) defined as:

$$DOP = \frac{I(\sigma+) - I(\sigma-)}{I(\sigma+) + I(\sigma-)} * 100\% \qquad (1)$$

where $I(\sigma+)$ and $I(\sigma-)$ are the peak intensities of left and right circular polarized components of the PL emission, respectively. The DOP for monolayer MoS$_2$, R-NEAPbI$_3$/MoS$_2$, and S-NEAPbI$_3$/MoS$_2$ are 0%, −7%, and 8%, respectively. A value of DOP of 0% obtained for monolayer MoS$_2$ confirms that artifacts due to difference in transmission of σ+ and σ− of optical components of the confocal setup, such as mirrors and lenses, are negligible and validates our measurements. In Supplementary Fig. 7, we show PL measured at different laser intensities for a set of R-NEAPbI$_3$/MoS$_2$ and S-NEAPbI$_3$/MoS$_2$ heterostructures which also showed DOP of −5 ± 1% and 6 ± 2%, respectively. Heterostructures transferred on hBN also demonstrated a DOP of −6%, confirming that the substrate does not affect the DOP (Supplementary Fig. 8). Time-dependent PL measurements (Supplementary Figs. 9, 10) show that the observed phenomenon is also stable over time.

By using circularly polarized excitation we could further enhance the DOP. Figure 3 shows PL measurements performed with circularly polarized 1.96 eV laser. For MoS$_2$ monolayer, we obtained DOP of −3% and 3% for excitation with σ− and σ+, respectively. The DOP of R-NEAPbI$_3$/MoS$_2$ heterostructure was −11% for σ− excitation and 0% for σ+ excitation. Similarly, the DOP of S-NEAPbI$_3$/MoS$_2$ heterostructure was 0% and 13% for σ− and σ+ excitation, respectively. These results are also consistent with spin selective electron transfer from MoS$_2$ to chiral perovskite. In bare monolayer MoS$_2$, the emission is higher when the polarization of excitation and detection are the same (σ+σ+ or σ−σ−) in comparison to the case when they are opposite (σ−σ+ or σ+σ−), as we pump and probe the same valley in the former case[4–8]. The DOP for σ+ and σ− excitation also have the same magnitude but opposite sign since the two valleys are symmetric. In R-NEAPbI$_3$/MoS$_2$ heterostructure, however, when K′ valley

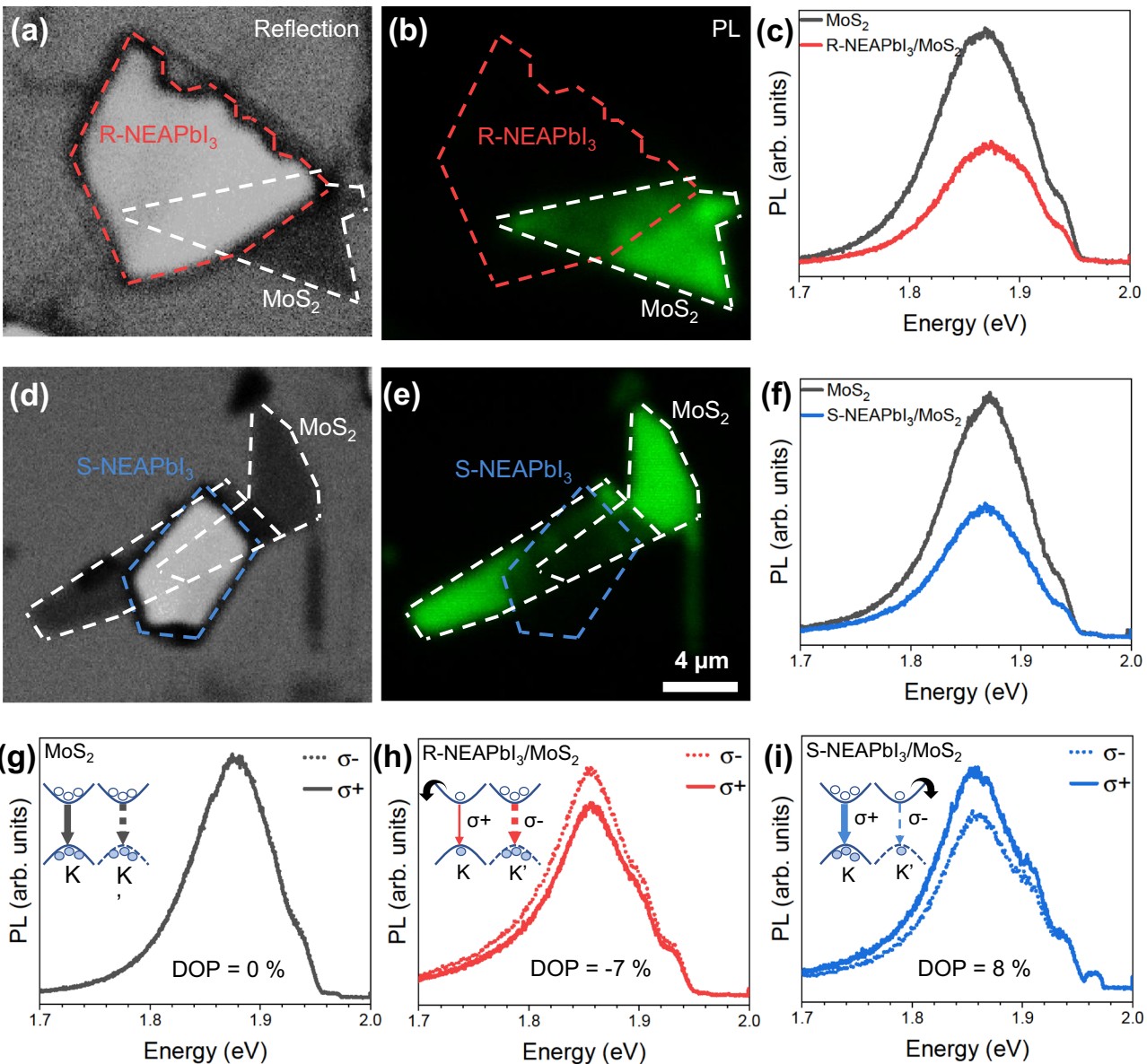

**Fig. 2 | Photoluminescence of MoS₂/1D chiral perovskite heterostructures.**
**a** Optical reflection image and **b** PL map of R-NEAPbI₃/MoS₂ heterostructure. **c** PL spectra from MoS₂ only region (black) and overlap region (red) of R-NEAPbI₃/MoS₂ heterostructure. **d** Optical reflection image and **e** PL map of S-NEAPbI₃/MoS₂ heterostructure. **f** PL spectra from monolayer MoS₂ only region (black) and overlap region (blue) of S-NEAPbI₃/MoS₂ heterostructure. The dotted white, red, and blue lines outline the monolayer MoS₂, R-NEAPbI₃ flake and S-NEAPbI₃ flake, respectively. Polarization resolved PL spectra of **g** monolayer MoS₂, **h** R-NEAPbI₃/MoS₂, and **i** S-NEAPbI₃/MoS₂. The solid lines and dotted lines show left (σ+) and right (σ−) circularly polarized components of the PL, respectively. The inset shows schematics of electronic bands at K and K' valley with respective optical selection rules. The arrows indicate preferential electron transfer from specific valleys in MoS₂ to R- and S-NEAPbI₃ chiral perovskite, respectively. Electrons and holes are depicted as empty and filled circles.

is selectively excited by σ−, σ+ emission from K valley is further reduced due to preferential transfer of spin up electrons from K valley to R-NEAPbI₃. Therefore, the *DOP* for σ−(σ+) excitation is further enhanced while that for σ+(σ−) excitation is diminished in R-NEAPbI₃(S-NEAPbI₃)/MoS₂ heterostructure in comparison to bare MoS₂ monolayer. We note that besides the peak at 1.86 eV, the emission appears to have an additional peak at 1.89 eV. As we observed this signal also from SiO₂ substrate measured under the same conditions, we attribute the peak to scattered laser leaking in through the filters (Supplementary Figs. 11, 12).

The energy of optical excitation is another parameter that can be used to manipulate valley polarization in these heterostructures. Figure 4 shows polarization resolved PL spectra of the R-NEAPbI₃/MoS₂ heterostructure obtained with linear polarized

laser excitation of 1.96 eV, 2.09 eV, and 2.33 eV in Fig. 4a–c, respectively. For the A-exciton transition, we observed a *DOP* of −7% for the 1.96 eV laser excitation which decreased to −3% for the 2.09 eV optical excitation and for the same heterostructure sample. Similarly, no polarization, i.e., a *DOP* = 0%, was observed for the 2.33 eV optical excitation. These are observations consistent with reports pointing to a lower valley polarization under non-resonant excitation conditions due to higher intervalley scattering[5,35]. We note that for 2.09 eV optical excitation, the noise from the scattered laser made it difficult to access the polarization of the B-exciton emission located at about 2.02 eV. However, for R-NEAPbI₃/MoS₂, we expect to observe an opposite polarization behavior for the B-exciton with 2.09 eV excitation, i.e., *DOP* values negative and positive for A- and B-exciton

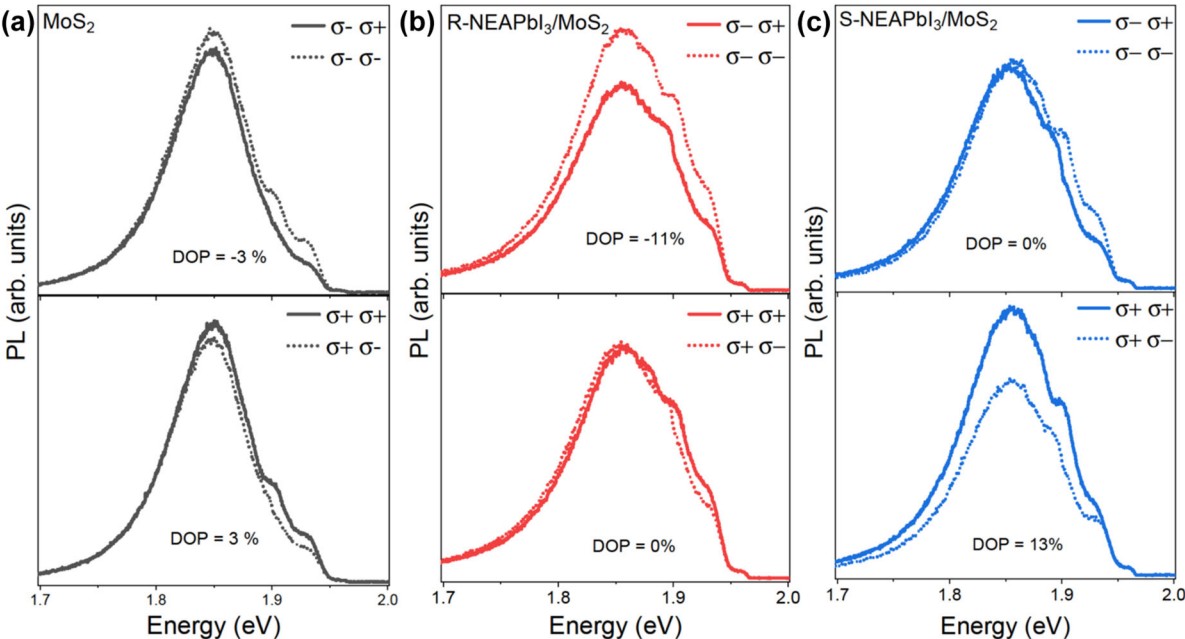

**Fig. 3 | Photoluminescence with circularly polarized excitation.** Polarization resolved PL spectra with circularly polarized excitation of **a** monolayer $MoS_2$, **b** R-NEAPbI$_3$/MoS$_2$, and **c** S-NEAPbI$_3$/MoS$_2$. The legend indicates the polarization of excitation followed by polarization of the emission. The top and bottom plots show measurements performed with right (σ−) and left (σ+) circularly polarized excitation, respectively, at 1.94 eV. The solid lines and dotted lines show σ+ and σ− circularly polarized components of the emitted PL, respectively.

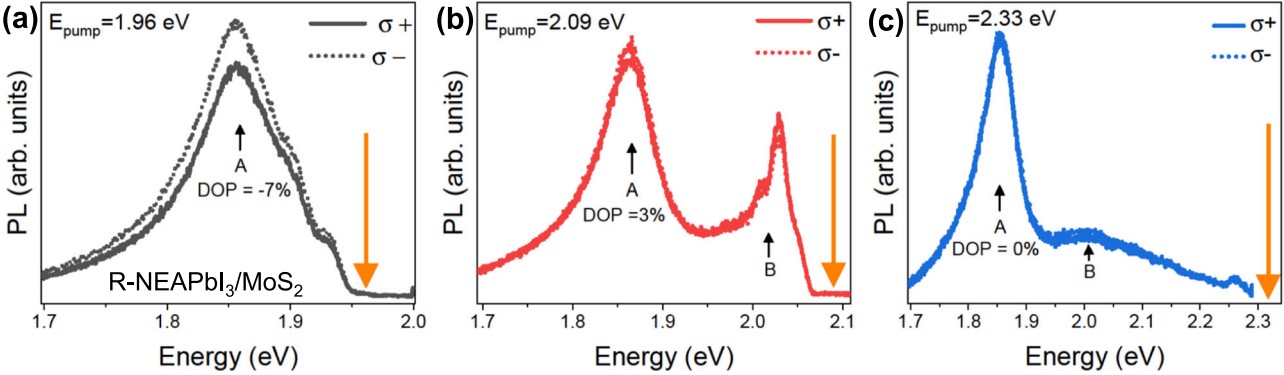

**Fig. 4 | Laser excitation energy dependency of valley properties.** Helicity resolved PL spectra of R-NEAPbI$_3$/MoS$_2$ with **a** 1.96 eV (or 633 nm), **b** 2.09 eV (or 594 nm), and **c** 2.33 eV (or 532 nm) laser excitation. The solid and dotted lines show left (σ+) and right (σ−) circularly polarized components of the PL, respectively. The orange arrows indicate the energy of the laser used for optical excitation.

transitions, respectively. This is because a linearly polarized laser with an energy of 2.09 eV will excite electrons from both the upper and lower spin split valence bands, while only spin up electrons are preferentially transferred from the conduction band of monolayer $MoS_2$ to R-NEAPbI$_3$ in the R-NEAPbI$_3$/MoS$_2$ heterostructure. Due to valley contrasting spin splitting of valence bands, the resulting population imbalance in the K and K′ valleys should be opposite for A and B excitons (schematic in Supplementary Fig. 13). Therefore, by using chiral perovskites as spin filters, it is possible to simultaneously populate both the K and K′ valleys with either spin up or spin down electrons. With a circularly polarized excitation, one can manipulate only a specific valley: irrespective of the energy of the optical excitation, σ+(σ−) will populate only the K (K′) valley. However, the spin filtering strategy offers the possibility of using excitation energy as an additional parameter to manipulate valley and spin polarization.

**Transient absorption spectroscopy measurements**

We obtained direct evidence for a photoinduced charge transfer in the presented heterostructures by pump-probe transient absorption spectroscopy. We optically pumped the samples at 2.48 eV (40 μJ cm$^{-2}$ fluence, 200 fs pulse width) and measured the photo-induced changes in absorption of a white light continuum probe after a controlled time delay. Figure 5a shows the differential transmission 2D plot (top) and transient spectra at 0.3 ps delay (bottom) of the MoS$_2$/S-NEAPbI$_3$ heterostructure, with features at 1.8 and 2.0 eV corresponding to the A- and B-exciton transitions of MoS$_2$, respectively[36–38] (Supplementary Fig. 14). The decay of the A-exciton at 1.87 eV for the heterostructure versus the monolayer $MoS_2$ is shown in Fig. 5b. Using a single exponential decay model (solid lines), we obtained carrier lifetimes of 2.2 and 1.6 ps for the monolayer $MoS_2$ ($\tau_{mono}$) and for the heterostructure MoS$_2$/S-NEAPbI$_3$ ($\tau_{Het}$), respectively. We attribute the

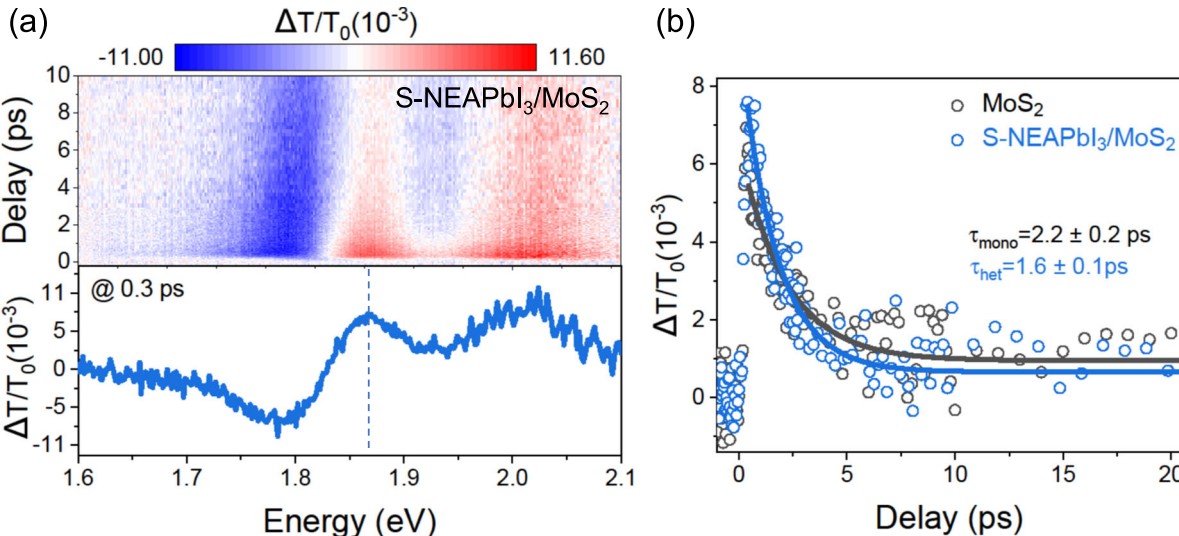

**Fig. 5 | Broadband transient absorption spectra of S-NEAPbI₃/MoS₂ heterostructure. a** 2D image of a differential transmission spectra ($\Delta T/T_0$) as a function of probe delay. The color bar shows the intensity of the $\Delta T/T_0$ with red and blue indicating positive and negative values, respectively. The plot at the bottom shows $\Delta T/T_0$ spectra at a delay of 0.3 ps. **b** Decay kinetics of monolayer MoS₂ (black circles) and S-NEAPbI₃/MoS₂ heterostructure (blue circles) at 1.87 eV (shown by a dotted line in (**a**)). The solid lines are fit to mono-exponential decay function.

shorter carrier lifetime observed in the heterostructure to electron transfer from monolayer MoS₂ to 1D chiral perovskite. Faster decays due to ultrafast charge transfer between TMDs and perovskites (as fast as 45 fs) have been previously reported[39–43] from transient absorption and time-resolved PL measurements. We estimated a charge transfer rate, $1/\tau_{CT} = 1/\tau_{Het} - 1/\tau_{mono}$, and a charge transfer efficiency, $\Phi_{CT} = 1 - \tau_{Het}/\tau_{mono}$, of $1/5.9$ ps⁻¹ and 27%[44,45].

## Discussion

In an ideal case, where there is good contact between the monolayer TMD/1D perovskite layers of the heterostructure and a 100% spin selectivity of the chiral perovskites, for a S-NEAPbI₃/MoS₂ heterostructure one expects that all the electrons from the conduction band in the K′ valley transfer to S-NEAPbI₃, while no electrons from the K valley transfer to S-NEAPbI₃. This ideal case would result in MoS₂ PL quenching of 50% and a *DOP* of 100%. Although we observed roughly 50% quenching, the *DOP* value of 8% is much lower. Possible reasons can be: (1) electrons from K′ valley can recombine with holes radiatively, before being transferred to S-NEAPbI₃; (2) electrons from K valley can be transferred to S-NEAPbI₃ since the spin selectivity of S-NEAPbI₃ is less than 100%; or (3) electrons from both K and K′ valleys can scatter to the opposite valley and depolarize. While depolarization due to intervalley scattering can be suppressed by lowering the temperature, requirements for cryogenic temperature would increase complexity and cost of the experiment. Alternative strategies can be adopted in the future to enable faster charge transfer and higher spin selectivity in such heterostructures. As hybrid lead halide perovskites offer large flexibility in terms of composition engineering, the energy band alignment in the heterostructure can be tuned to facilitate faster charge transfer. Similarly, more suitable chiral ligands could be designed to improve spin selectivity.

In summary, we demonstrate robust valley polarization in monolayer MoS₂/1D chiral perovskite heterostructures at room temperature via spin selective charge transfer. By pumping with a linearly polarized laser light close to A exciton resonance and measuring helicity resolved PL, we obtain a *DOP* of 0%, −7%, and 8% for monolayer MoS₂, MoS₂/R-, and S-NEAPbI₃ heterostructures, respectively. We attribute this to the chiral perovskites which act as a spin filter and control the valley population in monolayer MoS₂ via spin selective

charge transfer. We show that the polarization decreases under nonresonant excitation energy and predict that by selecting appropriate excitation energy, either spin up or spin down carriers could be selectively populated in both the valleys. Moreover, using ultrafast transient absorption spectroscopy, we present direct evidence for charge transfer from MoS₂ to chiral perovskite. Our results provide a simple and robust route to achieve valley polarization in monolayer TMDs without the need of low temperature, magnetic field, or complex device structure, thus paving the way for practical valleytronics devices.

## Methods
### Sample preparation

Chiral lead halide perovskite single crystal was synthesized following protocols reported in literature[21–23]. In brief, equimolar amounts of R- and S-1-(1-naphthyl)ethylamine (R- and S-NEA, from Tokyo Chemical Industries) and lead iodide (PbI from Sigma Aldrich) were dissolved in HI (47% in H₂O from Sigma Aldrich) and H₃PO₂ (50% in H₂O from Sigma Aldrich). The ratio of HI:H₃PO₂ was 5:1. The solution was heated to 100 °C under ambient conditions and continuous stirring. Once a clear solution was obtained, the solution was slowly cooled to room temperature resulting in single crystal precipitation. The single crystals were rinsed in chloroform and dried in vacuum at 60 °C for 3 h.

Perovskite thin films used for CD spectroscopy were made by spin coating inside an Argon filled glovebox. 10 wt% perovskite solution in Dimethylformamide (DMF) was prepared. 100 μl of the precursor solution was deposited on quartz substrates cleaned in acetone and isopropanol in an ultrasonic bath for 10 min each. The substrates were spun at 4000 rpm for 30 s. Finally, the substrates were annealed at 100 °C for 30 min.

To produce monolayer MoS₂, bulk MoS₂ crystals (from HQ Graphene) were mechanically exfoliated using the standard "scotch tape" method. In brief, a small piece of bulk MoS₂ was placed on a sticky tape and then the tape was folded a peeled several times to thin down the flake. Finally, the tape was pressed onto a clean Si substrate with 290 nm thick SiO₂ and slowly peeled off. Monolayers transferred on the Si substrate were identified with an optical microscope and confirmed with Raman micro-spectroscopy and AFM.

For transmission measurements, a CVD grown monolayer of MoS₂ on sapphire substrate (from 2D Semiconductors) was used.

Heterostructures were made with dry transfer technique[31] using a commercial transfer stage from HQ graphene under ambient conditions. Perovskite flakes were directly exfoliated on a Poly-dimethylsiloxane (PDMS) stamp (from Gelpak). The PDMS stamp with perovskite was turned upside down and aligned on top of a monolayer $MoS_2$. The PDMS stamp was slowly lowered and pressed on top of the target monolayer $MoS_2$ flake. After the entire perovskite flake was in contact with $MoS_2$, the PDMS stamp was slowly retrieved. This resulted in the perovskite flake being transferred from the PDMS stamp to the monolayer $MoS_2$ flake on Si/$SiO_2$ substrate. Perovskite flakes were transferred on CVD grown monolayer $MoS_2$ on sapphire substrate (2D Semiconductors) for transient absorption measurements. For all other measurements mechanically exfoliated monolayer $MoS_2$ on Si/$SiO_2$ substrate were used.

## Characterization

Photoluminescence (PL) was measured with a home-built confocal microscope consisting of an Olympus IX81 inverted microscope equipped with a ×60 dry objective lens, NA 0.7 (Olympus America) and delivering a spatial resolution of ~0.59 μm. Samples were excited with a continuous wave monochromatic laser light of 633 nm (He Ne Melles Griot), 594 nm (He Ne Melles Griot), or 532 nm (Diode laser Sapphire, Coherent). A polarizing beam-splitter cube was used to linearly polarize the excitation beam. Appropriate laser-line optical filters were used to clean the excitation laser. The PL emitted from the sample was collected by the same objective lens. A dichroic beam-splitter mirror and a long-pass filter were used to filter out the excitation laser from the emission. For PL spectra, the emission was directed to a monochromator (Acton SpectraPro 2750) via a multi-mode optical fiber. A quarter waveplate and a linear polarizer were added in front of the optical fiber to separately detect the left and the right circularly polarized PL emission for helicity resolved PL measurements. Similarly, another quarter waveplate was added after the linear polarizer to create circularly polarized laser excitation. For PL mapping, the emission was spatially filtered with a 100 μm pinhole. To simultaneously measure PL and reflection images, the emission was split by a 50:50 cube and imaged onto two single-photon-counting avalanche photodiodes (MPD, Picoquant). For the PL map, an additional long-pass filter was used to filter out the laser. The photodiodes were coupled to a time analyzer (PicoHarp 300, Pico-Quant) and the SymPhoTime 64 software (Picoquant) was used for data acquisition and analysis.

Transient Absorption spectroscopy (TAS) was performed with a commercial broadband transient absorption microspectrometer (Helios Fire, Ultrafast Systems). The fundamental optical output from an Ytterbium-doped Potassium-Gadolinium Tungstate (Yb:KGW) laser (PHAROS, Light conversion, 1030 nm, 195 fs, 1 kHz) was split into two parts. One part was directed to an optical parametric amplifier (Orpheus/Lyra, Light conversion) to generate a pump beam (500 nm). The other part was focused on a sapphire crystal to generate a super-continuum white-light probe beam. The time delay between the pump and probe beams was controlled using a computer-driven optical delay stage (SmartDelay Line, Ultrafast Systems). The beams were combined and focused on the sample with a ×20 objective lens (NA 0.6, Nikon). The laser spot of the pump and the probe were 10 μm and 7 μm in diameter, respectively. The transmitted probe beam was collected by another ×20 objective lens and directed to the Helios spectrometer, with experiments performed at a fluence of 40 μJcm⁻². All measurements were performed under ambient conditions.

Power X-ray diffraction (XRD) was performed with Rigaku Smar-tLab Universal Diffractometer. R-SNEAPbI$_3$ single crystals were crushed into fine powder using mortar and pestle. X-ray tube was operated at 40 kV and 45 mA.

Raman micro-spectroscopy was performed with a WiTec Alpha 300/Apron microscope (a part of the Quantum Materials Press facility

at the Center for Functional Nanomaterials). A 532 nm laser was used for excitation in combination with a 100×objective lens.

Atomic force microscopy (AFM) was performed with an Asylum Cypher AFM ES VRS in AC tapping mode.

Circular dichroism (CD) spectroscopy of perovskite thin films on quartz substrates was measured with a Jasco J-815 CD Spectro-polarimeter. Each sample was flipped and measured from both directions with the beam first hitting the (1) perovskite side (front), and (2) the quartz side (back). The average of the front and the back measurements were computed to obtain CD spectra.

Ultraviolet photoelectron spectroscopy (UPS) was carried out in an ultrahigh vacuum system with base pressure of $2 \times 10^{-10}$ Torr equipped with a SPECS Phoibos 100, MCD-5 hemispherical energy analyzer, and an ultraviolet source UVS 10/35. He(I) at 21.2 eV radiation was used. The angle between the analyzer and UV- source was 45° and photoelectrons were collected along the sample surface normal. R-SNEAPbI$_3$ single crystals were crushed into fine powder using mortar and pestle and placed on a Flag style sample plate. The spectra were calibrated vs the Fermi edge of Au (111) crystal. The work-function of the material was determined by subtracting the ultraviolet radiation energy (21.2 eV) from the secondary electron cut-off energy of UPS. The energy gap between valence band maxima to Fermi level was determined by UPS low binding energy edge to Fermi level. Bandgap between conduction and valence bands was obtained by UV-vis. Thus, the conduction band (CB) edge, valence band (VB) edge and Fermi level relative to the vacuum level was estimated as shown in Fig. 1d. Full data in SI Fig. 4.

UV-vis transmission of monolayer $MoS_2$ was measured in an inverted microscope (Nikon Eclipse Ti2, New Technology & Consulting, Germany). The samples were illuminated with a broadband light source (Energetiq LDLS EQ-99) and the transmitted light was directed to a monochromator (Acton SpectraPro HRS-500) and a PIXIS Roper camera.

## Data availability

All data generated and analyzed during this study are included in this article and its supplementary information files and are available on reasonable request.

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

## Acknowledgements

This research used facilities at the Center for Functional Nanomaterials (CFN), which is a U.S. Department of Energy Office of Science User Facility, at Brookhaven National Laboratory under Contract No. DE-SC0012704. S.S., M.L., S.P., X.T., and M.C. are supported by U.S. Department of Energy Office of Science Grant DE-SC0012704. The authors thank Xuance Jiang from Stony Brook University for discussions and Mr. Michael Maikowski of Coherent for loaning a Sapphire diode laser.

## Author contributions

S.S. and M.C. conceived the idea, S.S. prepared the samples with help from M.L. and S.P., S.S. and M.L. performed the measurements under the supervision of M.C., X.T. performed UPS measurements, S.S. pre-pared the manuscript with M.L., D.D., and M.C. All authors contributed to discussion of results in the manuscript.

## Competing interests

The authors declare no competing interests.
