## [Peer Review File · Nature Communications]

Reviewers' Comments:

Reviewer #1:

Remarks to the Author:

In this manuscript, the authors demonstrate that the valley polarization of MoS₂ can be tuned at room temperature by spin injection from one-dimensional chiral perovskite via stacking them to form type-II band alignment. Overall, this study is important to the community. Therefore, I would like to recommend its publication after the authors successfully addressed my following concerns.

1 Is it possible to provide circularly polarized light emission from the as-synthesized 1D chiral perovskites?

2 1D chiral perovskite crystals are expected to exhibit optical anisotropy, which might affect the accuracy of circularly polarized emission. Can the authors comment on this?

3 Can the authors explain why they can achieve valley modulation at room temperature?

According to previous report in MoS₂/2D chiral perovskite heterostructures, it is impossible to achieve valley modulation at room temperature due to the spin flipping.

4 Can the authors explain why the emission of MoS₂ has not been completely quenched?

Normally, it is expected that the charge transfer should be rather efficient so that the emission of monolayer MoS₂ should be totally quenched.

5 In type-II band alignment, the interlayer excitons are expected to be formed, which has been reported in 2D perovskite/monolayer TMD (<https://doi.org/10.1021/acs.nanolett.2c02742>). Can the authors comment on this?

6 In Figure 2, it seems that the emission peak of MoS₂ and also of heterostructure are rather broad and there are several peaks. Can the authors explain this?

7 In Figure 3b, under the 2.09 eV laser excitation, the A-exciton emission of MoS₂ also exhibit a considerable valley polarization. How about B-exciton emission? Can the authors provide more experimental data under different laser excitation with energy between 1.96 eV and 2.33 eV so that the evolution of DOC with excitation energy can be clearly seen.

8 How about the long-term stability of the heterostructures?

Reviewer #2:

Remarks to the Author:

The manuscript by Mircea Cotlet and co authors describes optical studies of monolayer MoS₂ and naphthylethylamine based one-dimensional chiral lead halide perovskite. The results of steady state PL and time resolved TA measurements are interpreted as efficient charge transfer between two materials due to type 2 band alignment as well as authors demonstrate room temperature valley polarization in the emission from MoS₂ originating from chiral character of perovskites. Despite the measurements are performed correctly, their interpretation is not fully convincing in the current version of the manuscript. Several points should be clarified and improved prior to any publications:

1. First of all what is uniformity of the samples. The PL and reflectivity mapping experiments should be performed to really quantify reproducibility of the data.

2. The real PLE experiment should be performed with reflectivity spectrum overlaid on the top to really identify the resonances.

3. The spectra of the emission of MoS₂ are really composed of multiple peaks as well as are very broad, which suggest a very low quality of the material. If there are many defects this might dramatically influence the results as well as interpretation. One simple solution is to prepare samples encapsulated in h-BN to improve optical quality. The reflectivity spectrum should be added taken on bare MoS₂ and HS. What is the origin on such a multiple peak structure on MoS₂? Clearly it is not fully excitonic and the interpretation is based on excitonic picture.

4. Authors should provide full band structure. At the moment we can learn from manuscript that we have type 2 band alignment, but it is schematically presented only for conduction band. What are the band offsets in the valence band?

5. If we have efficient charge transfer of the electron to perovskites and the hole is staying in MoS₂ than we should see formation of the interlayer (indirect) exciton. The emission at low energy range should be investigated. and the circular emission of such interlayer exciton would be much better support for the interpretation of the data.

6. Another test, which should be performed is low temperature spectroscopy. If there is really transfer of the electron from MoS₂ than charge exciton population will be dramatically impact and it will be observed in the emission and absorption spectra out and on HS.

7. Did authors try to excite with circular polarization?

Point by point response to reviewers

We thank the reviewers for taking their time to carefully assess our work and for their constructive feedback. We have addressed their concerns point-by-point as detailed below. Reviewer comments are shown in blue for easy distinguishing from our responses. Changes to the revised manuscript and supporting information can be seen in the highlighted copies that we have attached to the submitted materials.

Reviewer #1:

In this manuscript, the authors demonstrate that the valley polarization of MoS₂ can be tuned at room temperature by spin injection from one-dimensional chiral perovskite via stacking them to form type-II band alignment. Overall, this study is important to the community. Therefore, I would like to recommend its publication after the authors successfully addressed my following concerns.

We thank reviewer #1 for his/her feedback and comments and the encouraging recommendation. We hope we have addressed all his/her concerns in the revised manuscript.

1. Is it possible to provide circularly polarized light emission from the as-synthesized 1D chiral perovskites?

The 1D chiral perovskite has exciton absorption peak at 3.3 eV so we expect the emission to be in the UV. Unfortunately, due to limitations of the spectral range of our confocal micro photoluminescence system and of the polarization optics (quarter waveplate, linear polarizers), we could not measure the circularly polarized light emission from the as synthesized 1D chiral perovskites.

2. 1D chiral perovskite crystals are expected to exhibit optical anisotropy, which might affect the accuracy of circularly polarized emission. Can the authors comment on this?

The chiral perovskites crystals we used here have a 1D molecular structure (PbI chains surrounded by chiral ligands). They do not have a 1D morphological structure like in nanowires so we do not necessarily expect their shape to affect the accuracy of circularly polarized emission. Typically, the perovskite flakes we used are about 500 nm thick and have a maximum lateral dimension of 10-20 microns.

These 1D chiral perovskites have a wide bandgap (3.3 eV) that we do not expect to feature an emission overlapping with MoS₂ emission which is located at about 1.8 eV. The steady state absorption spectrum of the 1D chiral perovskite we show in Figure R1.a (also in SI, Fig. 2, revised manuscript) features an exciton absorption peak at 375 nm, with an edge at around 400 nm. In Figure R1.b, we show the PL from SiO₂ substrate (black), 1D chiral perovskite S-NEAPbI₃ (red) on SiO₂ substrate and MoS₂/chiral perovskite heterostructure on SiO₂ substrate (blue) obtained with 2.02 eV excitation. We can see that the signal from perovskite is negligible, containing only a small peak from laser scattering (2.02 eV) which can also be observed for the SiO₂ substrate. Therefore, we did not observe any absorption or emission from

perovskites to be present in the spectral range of MoS₂ emission. Therefore, we do not expect that the anisotropy of perovskite to affect the accuracy of the reported circular polarized measurements.

Figure R1. (a) Absorption of S-NEAPbI₃. (b) PL signals measured from SiO₂ substrate (black), 1D chiral perovskite SNEAPbI₃ on SiO₂ substrate (red) and MoS₂/chiral perovskite heterostructure (blue) on SiO₂ substrate with 2.02 eV optical excitation.

3. Can the authors explain why they can achieve valley modulation at room temperature? According to previous report in MoS₂/2D chiral perovskite heterostructures, it is impossible to achieve valley modulation at room temperature due to the spin flipping.

In the previous report on MoS₂/2D chiral perovskite the authors observed about 12% DOP at 78 K, which gradually decreased as the temperature increased, to become negligible at room temperature.¹ They attributed this observations to spin flipping, intervalley scattering and lower DOP of their 2D chiral perovskite at room temperature. Here, in our present work, we achieved valley modulation at room temperature because: (1) 1D chiral perovskites have a higher chiro-optical activity than 2D chiral perovskites.^{2,3} This could be because of stronger helical lattice distortions in the 1D PbI inorganic chains, as they are surrounded by chiral optical ligands from all sides in 1D perovskites. Whereas in 2D perovskites the chiral ligands lie only at the top and bottom of the 2D PbI layer. We expect the stronger chiral response to translate to stronger spin selection; (2) In the previous report they excited the sample using 2.33 eV which is higher than the A exciton resonance (1.86 eV). When photoexcited carriers have excess energy, there is a higher probability of spin flipping or intervalley scattering and this has been reported in literature.⁴ In our work we excite the sample with 1.96 eV, which is closer to the A exciton resonance.

4. Can the authors explain why the emission of MoS₂ has not been completely quenched? Normally, it is expected that the charge transfer should be rather efficient so that the emission of monolayer MoS₂ should be totally quenched.

We think of two possible reasons why the emission of MoS₂ is not completely quenched: (1) the interface between MoS₂ and chiral perovskite may not be homogeneous, and this can be due to possible contaminants, polymer residue during transfer and air pockets. Although we expect strong coupling between monolayer MoS₂ and perovskite in the majority of the overlap region, some MoS₂ monolayer from small regions may not have good contact with the perovskite and this can contribute to PL. This has been reported previously in TMD/TMD heterostructures where a cleaner interfaces is expected that in the present case.⁵ (2) in the ideal case, with a strong interface coupling and 100% spin selectivity, only electrons with one spin would be transferred. Therefore, only one valley would be quenched completely while the opposite valley would not be quenched, resulting in 50% overall quenching. Although in our case, the spin selectivity is very likely less than 100%, we still expect preferential quenching from only of the two valley and incomplete quenching from the opposite valley.

5. In type-II band alignment, the interlayer excitons are expected to be formed, which has been reported in 2D perovskite/monolayer TMD (<https://doi.org/10.1021/acs.nanolett.2c02742>). Can the authors comment on this?

We also expected to see emission from interlayer excitons. From UPS and UV-vis absorption measurements we obtained a conduction band minima for the 1D chiral perovskite of -4.4 eV. The valence band maxima of MoS₂ monolayer (according to literature reports) ranges from about -5.87 eV to -5.77 eV.^{6,7} Therefore, we expected the interlayer exciton to emit at around 1.37 eV to 1.47 eV. We measured spectra from 1.3 eV to 2.0 eV and did not observe any additional emission from interlayer excitons in this range (Figure R2). We speculate that interlayer excitons may emit at energies below 1.3 eV which are in the IR region, where the quantum efficiency of our detector (PIXIS 100B, Princeton Instruments) is less than 30 %, as opposed to more than 95 % at 2.0 eV.⁸ Another possibility is that interlayer exciton emission is weak or dark altogether, a situation which has also been reported in literature.^{1,9,10}

We have added the following discussion paragraph in our revised manuscript:

“Emission at low energy from interlayer excitons, which have been reported in TMD/2D perovskite heterostructures^{11,12}, was absent in the spectral range probed (1.3 eV – 2 eV). Therefore, in this work we focus on only MoS₂ emission.”

Figure R2. PL spectra of MoS₂/R-NEAPbI₃ heterostructures

6. In Figure 2, it seems that the emission peak of MoS₂ and also of heterostructure are rather broad and there are several peaks. Can the authors explain this?

We understand the reviewer's concern. The FWHM of the peaks from MoS₂ and heterostructure were between 80 meV to 100 meV. As Reviewer#2 suggested, we transferred the MoS₂ exfoliated on Si/SiO₂ substrate to hBN and made hBN/MoS₂/R-NEAPbI₃ heterostructure. These heterostructures had a much narrower FWHM of 56 meV (Figure R3.a) which is similar to values reported in literature at room temperature.^{13,14} As the same MoS₂ bulk crystal was used for exfoliation on SiO₂ and the samples were stored in the same condition, we believe that the broad emission is due to impurities and roughness of SiO₂ surface.

Besides the main peak, the emission appears to have an additional peak at 1.89 eV. This signal was also observed from SiO₂ substrates measured under the same conditions (Figure R3b). Therefore, we attribute the peak to scattered laser leaking in through the dichroic filter and long pass filter. The shoulder like feature at 1.93 eV is most probably from the optical filter edge. This is further supported by data from heterostructures on hBN (Figure R3a). In this case, since the emission from the sample was high, the peak at 1.89 eV is absent while a slight shoulder can still be seen at 1.93 eV.

Figure R3. PL of MoS₂/R-NEAPbI₃ heterostructure on (a) hBN (b) laser reflection from SiO₂

More importantly, we obtained a similar DOP of -6.5 % from hBN/MoS₂/R-NEAPbI₃ heterostructure (Figure R4). This indicates that the broad spectra of MoS₂ on SiO₂ substrate (possibly due to impurities and roughness of SiO₂ surface) and multiple peaks (due to measurement artifacts) do not influence our results and interpretation.

Figure R4. Helicity dependent PL spectra of hBN/MoS₂/R-NEAPbI₃

We have added the following paragraph in the revised manuscript:

“We note that besides the peak at 1.86 eV, the emission appears to have an additional peak at 1.89 eV. As we observed this signal also from SiO₂ substrate measured under the same conditions, we attribute the peak to scattered laser leaking in through the filters (Supplementary Figs. 11 and 12).”

We have included Figure R3 and R4 in SI as well, now supplementary Figs.11 and 12.

7. In Figure 3b, under the 2.09 eV laser excitation, the A-exciton emission of MoS₂ also exhibit a considerable valley polarization. How about B-exciton emission? Can the authors provide more experimental data under different laser excitation with energy between 1.96 eV and 2.33 eV so that the evolution of DOC with excitation energy can be clearly seen.

Under 2.09 eV, we observed 3% valley polarization from the A exciton emission. However, the case was not clear for the B exciton. The 2.09 eV laser excitation is close to the B exciton peak which is at about 2.01 eV. Therefore, the edge of the 592 nm long pass filter (used to filter out the 591 nm (2.09 eV) laser) also cuts a part of the B exciton emission and lets the 2.09 eV laser go through and as such we did not get conclusive results.

We also performed new measurements with 2.21 eV (561nm) laser excitation as the reviewer suggested. As shown in Figure R5, the difference between $\sigma+$ and $\sigma-$ for both A and B exciton emissions is negligible. We expect the DOP to decrease as the laser excitation energy moves further away from the exciton resonance peak.¹⁵ For A exciton emission, with 2.21 eV excitation we expect a DOP between that from 2.09 eV and 2.33 eV excitations, that is, between -3% and 0%. Since the limitation of our setup is about $\pm 1\%$ (due to possible distortion of the circularly polarized emission by lenses, mirrors, dichroic, etc) we could not get conclusive evidence.

Figure R5. Helicity resolved PL from R-NEAPbI₃/MoS₂ heterostructure with 2.21 eV excitation.

8. How about the long-term stability of the heterostructures?

Typically, we measured fresh samples within 3-4 days. We measured the samples under ambient conditions and stored the samples in a vacuum desiccator overnight and did not observe significant change in emission. However, after about a week we noticed that the DOP decreased from 7% to 6% (Figure R6). We think this is because of degradation of perovskite which is a known problem in organic-inorganic perovskites. As perovskites have ionic nature, they can be dissolved in polar solvents such as water. Therefore, degradation due to humidity is an issue. For perovskite solar cells, this has been overcome by using suitable encapsulation.¹⁶ We think similar strategies, for example hBN encapsulation, can be used in the future to improve the long-term stability of the heterostructures.

We do not think photodegradation of perovskite is a concern in our study since we do not optically excite perovskite and use very low laser intensity in our experiments (100 μ W).

Figure R6. Helicity resolved PL of MoS₂/R-NEAPbI₃ heterostructures (a) fresh (b) after one week

We have included Figure R6 in SI as a measure of sample stability for the reader.

Reviewer #2

The manuscript by Mircea Cotlet and co authors describes optical studies of monolayer MoS₂ and naphthylethylamine based one-dimensional chiral lead halide perovskite. The results of steady state PL and time resolved TA measurements are interpreted as efficient charge transfer between two materials due to type 2 band alignment as well as authors demonstrate room temperature valley polarization in the emission from MoS₂ originating from chiral character of perovskites. Despite the measurements are performed correctly, their interpretation is not fully convincing in the current version of the manuscript. Several points should be clarified and improved prior to any publications:

We thank Reviewer #2 for his/her feedback and comments. We hope the answers we provide here address all reviewer's concerns.

1. First of all what is uniformity of the samples. The PL and reflectivity mapping experiments should be performed to really quantify reproducibility of the data.

We agree with the reviewer that the uniformity of the sample is an important aspect. We show the reflectivity and PL mapping data for heterostructures in figure 2 a, b, d and e in the original manuscript and plot them here as Figure R7 for the MoS₂/R-NEAPbI₃ heterostructure. Although there is some variation in the PL intensity, we can clearly see that the overlap region of the heterostructure is darker than the monolayer MoS₂ region, providing mostly for uniform quenched PL in heterostructure.

Figure R7. Photoluminescence of MoS₂/1D chiral perovskite heterostructures (a) Optical reflection image and (b) PL map of R-NEAPbI₃/MoS₂ heterostructure. (d) Optical reflection image and (e) PL map of S-NEAPbI₃/MoS₂ heterostructure.

2. The real PLE experiment should be performed with reflectivity spectrum overlaid on the top to really identify the resonances.

Unfortunately, we could not get access to a supercontinuum laser to perform PLE measurements. However, we performed transmission microscopy measurements (Figure R8) and we can see that the MoS₂ A and B exciton resonances remain in the same positions in the heterostructures. As the chiral perovskite has a much wider bandgap of 3.3 eV, we do not expect the perovskite to absorb light emitted by MoS₂ A and B exciton which is at a lower energy than the absorption edge of the perovskite. The highest excitation energy we used in this work is 2.33 eV, which is also way below the absorption edge of perovskites.

Figure R8. Transmission of MoS₂ monolayer and S-NEAPbI₃/MoS₂ heterostructure

3. The spectra of the emission of MoS₂ are really composed of multiple peaks as well as are very broad, which suggest a very low quality of the material. If there are many defects this might dramatically influence the results as well as interpretation. One simple solution is to prepare samples encapsulated in h-BN to improve optical quality. The reflectivity spectrum should be added taken on bare MoS₂ and HS. What is the origin on such a multiple peak structure on MoS₂? Clearly it is not fully excitonic and the interpretation is based on excitonic picture.

We thank the reviewer for suggesting hBN encapsulation. The FWHM of the peaks from MoS₂ and heterostructure on SiO₂ were between 80 meV to 100 meV. We followed his/her recommendation and transferred the exfoliated MoS₂ onto hBN and made hBN/MoS₂/R-NEAPbI₃ heterostructures. These heterostructures had a much narrower FWHM of 56 meV (Figure R9.a) which is similar to values reported in literature at room temperature.^{13,14} As the same MoS₂ bulk crystal was used for exfoliation on SiO₂ substrate, we attributed the broad emission to impurities and roughness of SiO₂ surface.

Besides the main peak, the emission appears to have an additional peak at 1.89 eV. This signal was also observed from SiO₂ substrates measured under the same conditions (Figure R3b). Therefore, we attribute the peak to scattered laser light leaking through the dichroic filter and long pass filter. The shoulder-like feature at 1.93 eV is probably from the long pass filter edge. This is further supported by data from heterostructures on hBN (Figure R3a). In this case, since the emission from the sample was high in intensity and overwhelming the scattered laser, the peak at 1.89 eV is absent. While a slight shoulder can still be seen at 1.93 eV and related to the filter edge.

Figure R9. PL of MoS₂/R-NEAPbI₃ heterostructure on (a) hBN (b) SiO₂

More importantly, we obtained a similar DOP of -6.5 % from hBN/MoS₂/R-NEAPbI₃ heterostructure (Figure R10). This indicates that the broad spectra of MoS₂ on SiO₂ substrate (possibly due to impurities and roughness of SiO₂ surface) and multiple peaks (due to measurement artifacts) do not influence our results and interpretation.

Figure R10. Helicity dependent PL spectra of hBN/MoS₂/R-NEAPbI₃

We have added the following sentence in the revised manuscript:

“We note that besides the peak at 1.86 eV, the emission appears to have an additional peak at 1.89 eV. As we observed this signal also from SiO₂ substrate measured under the same conditions, we attribute the peak to scattered laser leaking in through the optical filters (Supplementary Fig. 11 and 12).”

We have included Figure R9 and R10 in SI.

4. Authors should provide full band structure. At the moment we can learn from manuscript that we have type 2 band alignment, but it is schematically presented only for conduction band. What are the band offsets in the valence band?

We experimentally determined the band structure of 1D chiral perovskite using UPS and UV-vis absorption and obtained a conduction band minima (CBM) of -4.4 eV and valence band maxima (VBM) of -7.7 eV. The CBM and VBM of MoS₂ monolayer according to literature reports are approximately -4.28 eV and -5.87 eV to -5.77 eV, respectively.^{6,7} Therefore, we expect a conduction band offset of 0.12 eV and valence band offset of 1.83-1.93 eV. The scale bar for energy in Figure 1d contains these values accordingly.

5. If we have efficient charge transfer of the electron to perovskites and the hole is staying in MoS₂ than we should see formation of the interlayer (indirect) exciton. The emission at low energy range should be investigated. and the circular emission of such interlayer exciton would be much better support for the interpretation of the data.

We agree with the reviewer on the formation of interlayer exciton which we expected to arise at around 1.37 eV to 1.47 eV. We measured spectra from 1.3 eV to 2.0 eV but did not observe any additional emission in this range (Figure R11). It could be that interlayer excitons emit at energies below 1.3 eV in the IR region, outside the range of our detector. Another possibility is that interlayer exciton emission is weak or dark altogether which has also been reported in literature.^{1,9,10}

Figure R11 PL spectra of MoS₂/R-NEAPbI₃ heterostructures

We have added the following discussion in the revised manuscript:

“Emission at low energy from interlayer excitons, which have been reported in TMD/2D perovskite heterostructures^{11,12}, was absent in the spectral range probed (1.3 eV – 2 eV). Therefore, in this work we focus on only MoS₂ emission.”

6. Another test, which should be performed is low temperature spectroscopy. If there is really transfer of the electron from MoS₂ than charge exciton population will be dramatically impact and it will be observed in the emission and absorption spectra out and on HS.

We agree that we could gain valuable insight from low temperature data. At low temperature we would expect higher DOP due to lower depolarization from intervalley scattering which was also reported earlier.¹ However, from transient absorption measurements we expect charge transfer to be faster than radiative recombination. Therefore, although temperature will affect the exciton population and hence the intensity of the PL signal, we may not necessarily see a difference in electron transfer. For TMD/TMD heterostructures, it has been previously reported that charge transfer is independent of temperature.¹⁹ However, here we limit our investigations to room temperature measurements. Our main message is that valley polarization can be achieved at room temperature. To support our charge transfer argument, we provide PL quenching and transient absorption data.

7. Did authors try to excite with circular polarization?

We performed new measurements with circularly polarized excitation as the reviewer suggested (Figure R12). The first σ symbol in the legend of this figure indicates the polarization of excitation and the second σ symbol indicates the polarization of the detected emission. For MoS₂ monolayer, we obtained a DOP of -3% and 3% for excitation with σ^- and σ^+ , respectively. The DOP of R-NEAPbI₃/MoS₂ heterostructure was -11% for σ^- excitation and almost 0% for σ^+ excitation. Similarly, the DOP of S-

NEAPbI₃/MoS₂ heterostructure was 0% and 13% for σ^- and σ^+ excitation, respectively. These results are also consistent with spin selective electrons transfer from MoS₂ to chiral perovskite. In R-NEAPbI₃/MoS₂ heterostructure for example, when K' valley is selectively excited by σ^- , the σ^+ emission from K valley is further reduced due to preferential transfer of spin up electrons from K valley to R-NEAPbI₃. Therefore, the DOP for σ^- (σ^+) excitation is enhanced while that for σ^+ (σ^-) excitation is diminished in R-NEAPbI₃ (S-NEAPbI₃)/ MoS₂ heterostructure in comparison to bare MoS₂ monolayer, where the DOP has the same magnitude but opposite sign for the two opposite circularly polarized excitations.

Figure R12. Photoluminescence with circularly polarized excitation. Polarization resolved PL spectra with circularly polarized excitation of (a) monolayer MoS₂, (b) R-NEAPbI₃/MoS₂ and (c) S-NEAPbI₃/MoS₂. The legend indicates the polarization of excitation followed by polarization of the emission. The top and bottom plots show measurements performed with σ^- and σ^+ circularly polarized excitation, respectively at 1.94 eV. The solid lines and dotted lines show σ^+ and σ^- circularly polarized components of the emitted PL, respectively.

The peak at 1.89 eV was also seen in SiO₂ (Figure R13).

Figure R13. Polarization resolved PL from S-NEAPbI₃/MoS₂ heterostructure and SiO₂ substrate.

To check the correctness of our measurement system, we measured a non-chiral reference sample (carboxyl quantum dots from Ocean NanoTech) with emission at 1.86 eV as reference for which we obtained a negligible DOP of less than $\pm 1\%$ (Figure R14).

Figure R14. Polarization resolved PL from a non-chiral reference (Carboxyl Quantum Dots from Ocean NanoTech)

We have added figure R12 and the following in the manuscript:

“By using circularly polarized excitation we could further enhance the DOP in heterostructure. Figure 3 shows PL measurements performed with circularly polarized 1.96 eV laser. For MoS₂ monolayer, we

obtained DOP of -3% and 3% for excitation with σ^- and σ^+ , respectively. The DOP of R-NEAPbI₃/MoS₂ heterostructure was -11% for σ^- excitation and 0% for σ^+ excitation. Similarly, the DOP of S-NEAPbI₃/MoS₂ heterostructure was 0% and 13% for σ^- and σ^+ excitation, respectively. These results are also consistent with spin selective electron transfer from MoS₂ to chiral perovskite. In bare monolayer MoS₂, the emission is higher when the polarization of excitation and detection is the same ($\sigma^+\sigma^+$ or $\sigma^-\sigma^-$) in comparison to when it is the opposite ($\sigma^-\sigma^+$ or $\sigma^+\sigma^-$), as we pump and probe the same valley in former case. The DOP for σ^+ and σ^- excitation also have the same magnitude but opposite sign since the two valleys are symmetric. In R-NEAPbI₃/MoS₂ heterostructure however, when K' valley is selectively excited by σ^- , σ^+ emission from K valley is further reduced due to preferential transfer of spin up electrons from K valley to R-NEAPbI₃. Therefore, the DOP for $\sigma^-(\sigma^+)$ excitation is further enhanced while that for $\sigma^+(\sigma^-)$ excitation is diminished in R-NEAPbI₃(S-NEAPbI₃)/MoS₂ heterostructure in comparison to bare MoS₂ monolayer. We note that besides the peak at 1.86 eV, the emission appears to have an additional peak at 1.89 eV. As we observed this signal also from SiO₂ substrate measured under the same conditions, we attribute the peak to scattered laser leaking in through the filters (see discussion in Supplementary Info). ”

We have added Figures R13 and R14 in the SI.

End of responses to all reviewers comments.

References

- (1) Chen, Y.; Ma, J.; Liu, Z.; Li, J.; Duan, X.; Li, D. Manipulation of Valley Pseudospin by Selective Spin Injection in Chiral Two-Dimensional Perovskite/Monolayer Transition Metal Dichalcogenide Heterostructures. *ACS Nano* **2020**, *14* (11), 15154-15160. DOI: 10.1021/acsnano.0c05343.
- (2) Ishii, A.; Miyasaka, T. Direct detection of circular polarized light in helical 1D perovskite-based photodiode. *Science Advances* **2020**, *6* (46), eabd3274. DOI: 10.1126/sciadv.abd3274.
- (3) Ma, J.; Wang, H.; Li, D. Recent Progress of Chiral Perovskites: Materials, Synthesis, and Properties. *Advanced Materials* **2021**. DOI: 10.1002/adma.202008785.
- (4) Kioseoglou, G.; Hanbicki, A. T.; Currie, M.; Friedman, A. L.; Gunlycke, D.; Jonker, B. T. Valley polarization and intervalley scattering in monolayer MoS₂. *Applied Physics Letters* **2012**, *101* (22). DOI: 10.1063/1.4768299.
- (5) Zhu, X.; Monahan, N. R.; Gong, Z.; Zhu, H.; Williams, K. W.; Nelson, C. A. Charge Transfer Excitons at van der Waals Interfaces. *J Am Chem Soc* **2015**, *137* (26), 8313-8320. DOI: 10.1021/jacs.5b03141 From NLM PubMed-not-MEDLINE.
- (6) Keyshar, K.; Berg, M.; Zhang, X.; Vajtai, R.; Gupta, G.; Chan, C. K.; Beechem, T. E.; Ajayan, P. M.; Mohite, A. D.; Ohta, T. Experimental Determination of the Ionization Energies of MoSe(2), WS(2), and MoS(2) on SiO(2) Using Photoemission Electron Microscopy. *ACS Nano* **2017**, *11* (8), 8223-8230. DOI: 10.1021/acsnano.7b03242 From NLM PubMed-not-MEDLINE.
- (7) Kang, J.; Tongay, S.; Zhou, J.; Li, J.; Wu, J. Band offsets and heterostructures of two-dimensional semiconductors. *Applied Physics Letters* **2013**, *102* (1), 012111. DOI: 10.1063/1.4774090 (accessed 2023/03/03).
- (8) Instruments, T. P. PIXIS: 100 eXcelon Rev. N5.1. https://www.princetoninstruments.com/wp-content/uploads/2020/05/PIXIS_100_datasheet-rebranded.pdf (accessed 2023 May 29).

- (9) Wang, K.; Huang, B.; Tian, M.; Ceballos, F.; Lin, M. W.; Mahjouri-Samani, M.; Boulesbaa, A.; Poretzky, A. A.; Rouleau, C. M.; Yoon, M.; et al. Interlayer Coupling in Twisted WSe₂/WS₂ Bilayer Heterostructures Revealed by Optical Spectroscopy. *ACS Nano* **2016**, *10* (7), 6612-6622. DOI: 10.1021/acsnano.6b01486.
- (10) Kozawa, D.; Carvalho, A.; Verzhbitskiy, I.; Giustiniano, F.; Miyauchi, Y.; Mouri, S.; Castro Neto, A. H.; Matsuda, K.; Eda, G. Evidence for Fast Interlayer Energy Transfer in MoSe₂/WS₂ Heterostructures. *Nano Lett* **2016**, *16* (7), 4087-4093. DOI: 10.1021/acs.nanolett.6b00801 From NLM PubMed-not-MEDLINE.
- (11) Yao, W.; Yang, D.; Chen, Y.; Hu, J.; Li, J.; Li, D. Layer-Number Engineered Momentum-Indirect Interlayer Excitons with Large Spectral Tunability. *Nano Lett* **2022**, *22* (17), 7230-7237. DOI: 10.1021/acs.nanolett.2c02742 From NLM PubMed-not-MEDLINE.
- (12) Chen, Y.; Liu, Z.; Li, J.; Cheng, X.; Ma, J.; Wang, H.; Li, D. Robust Interlayer Coupling in Two-Dimensional Perovskite/Monolayer Transition Metal Dichalcogenide Heterostructures. *ACS Nano* **2020**, *14* (8), 10258-10264. DOI: 10.1021/acsnano.0c03624.
- (13) Korn, T.; Heydrich, S.; Hirmer, M.; Schmutzler, J.; Schüller, C. Low-temperature photocarrier dynamics in monolayer MoS₂. *Applied Physics Letters* **2011**, *99* (10). DOI: 10.1063/1.3636402.
- (14) Lorchat, E.; Lopez, L. E. P.; Robert, C.; Lagarde, D.; Froehlicher, G.; Taniguchi, T.; Watanabe, K.; Marie, X.; Berciaud, S. Filtering the photoluminescence spectra of atomically thin semiconductors with graphene. *Nat Nanotechnol* **2020**, *15* (4), 283-288. DOI: 10.1038/s41565-020-0644-2 From NLM PubMed-not-MEDLINE.
- (15) Mak, K. F.; He, K.; Shan, J.; Heinz, T. F. Control of valley polarization in monolayer MoS₂ by optical helicity. *Nat Nanotechnol* **2012**, *7* (8), 494-498. DOI: 10.1038/nnano.2012.96.
- (16) Emery, Q.; Remec, M.; Paramasivam, G.; Janke, S.; Dagar, J.; Ulbrich, C.; Schlatmann, R.; Stannowski, B.; Unger, E.; Khenkin, M. Encapsulation and Outdoor Testing of Perovskite Solar Cells: Comparing Industrially Relevant Process with a Simplified Lab Procedure. *ACS Appl Mater Interfaces* **2022**, *14* (4), 5159-5167. DOI: 10.1021/acсами.1c14720 From NLM PubMed-not-MEDLINE.
- (17) Rivera, P.; Schaibley, J. R.; Jones, A. M.; Ross, J. S.; Wu, S.; Aivazian, G.; Klement, P.; Seyler, K.; Clark, G.; Ghimire, N. J.; et al. Observation of long-lived interlayer excitons in monolayer MoSe₂-WSe₂ heterostructures. *Nat Commun* **2015**, *6*, 6242. DOI: 10.1038/ncomms7242.
- (18) Zhang, Q.; Linaryd, E.; Wang, X.; Eda, G. Excitonic Energy Transfer in Heterostructures of Quasi-2D Perovskite and Monolayer WS₂. *ACS Nano* **2020**, *14* (9), 11482-11489. DOI: 10.1021/acsnano.0c03893.
- (19) Wang, Z.; Altmann, P.; Gadermaier, C.; Yang, Y.; Li, W.; Ghirardini, L.; Trovatiello, C.; Finazzi, M.; Duo, L.; Celebrano, M.; et al. Phonon-Mediated Interlayer Charge Separation and Recombination in a MoSe₂/WSe₂ Heterostructure. *Nano Lett* **2021**, *21* (5), 2165-2173. DOI: 10.1021/acs.nanolett.0c04955 From NLM PubMed-not-MEDLINE.

Reviewers' Comments:

Reviewer #1:

Remarks to the Author:

The authors have properly addressed my all questions and I would like to recommend its publication at the current form.

Reviewer #2:

Remarks to the Author:

I would like to thank the authors for their detailed response to the questions and comments raised by referees. I recommend the manuscript for publication.